# Modelling Functional Thyroid Follicular Structures Using P19 Embryonal Carcinoma Cells

**DOI:** 10.3390/cells13221844

**Published:** 2024-11-07

**Authors:** Fatimah Najjar, Liming Milbauer, Chin-Wen Wei, Thomas Lerdall, Li-Na Wei

**Affiliations:** Department of Pharmacology, University of Minnesota, Minneapolis, MN 55455, USA; najja022@umn.edu (F.N.); chang032@umn.edu (L.M.); wei00170@umn.edu (C.-W.W.); lerda013@umn.edu (T.L.)

**Keywords:** thyrocytes, embryonal carcinoma stem cells, cell differentiation, thyroid diseases, in vitro models

## Abstract

Thyroid gland diseases remain clinical challenges due to the lack of reliable in vitro models to examine molecular pathways of thyrocytes development, maturation, and functional maintenance. This study aimed to develop in vitro thyrocytes model using a stem cell culture, P19 embryonal carcinoma which requires no feeder layer, differentiation into mature and functional thyrocytes that allow molecular and genetic manipulation for studying thyroid diseases. The procedure utilizes Activin A and thyroid stimulating hormone (TSH) to first induce embryoid body endoderm formation enriched in thyrocyte progenitors. Following dissociating embryoid bodies, thyrocyte progenitors are plated in Matrigel as monolayer cultures that allows thyrocyte progenitors mature to functional thyrocytes. These thyrocytes further maturate to form follicle-like structures expressing and accumulating thyroglobulin that can be secreted into the medium upon TSH stimulation. Thyrocyte differentiation-maturation process is monitored by the expression of essential transcriptional factors and thyrocyte-specific functional genes. Further, the applicability of this system is validated by introducing a siRNA control. Following molecular manipulation, the system can still be guided to differentiate into mature and functional thyrocytes. This system spans a time frame of 14 days, suitable for detailed molecular studies to dissect pathways and molecular players in thyrocytes development and functional maintenance.

## 1. Introduction

Thyroid diseases remain a challenging clinical issue due to the lack of reliable in vitro models for understanding the physiological processes of thyroid gland development, maturation, and/or the maintenance of a healthy thyroid gland [1,2]. One of our recent studies has revealed a critical new functional role for a specific protein abundantly expressed in the thyroid gland, cellular retinoic acid binding protein 1 (CRABP1) [3]. The *Crabp1* gene knockout (CKO) mice exhibited adult-onset hypothyroidism due to increasingly defected thyrocytes starting from young adult stages [4], suggesting defects in thyrocyte formation, maturation, or functional maintenance. The need for a suitable in vitro experimental model prompted this study, aiming to develop an efficient and reliable in vitro model system for systemic examinations of molecular events and mechanisms leading to thyroid dysfunction.

Thyroid gland contains mainly follicular cells and parafollicular cells. Follicular cells are known as thyrocytes, which are the most abundant cells in the thyroid gland that produce and secrete thyroid hormones [5,6,7]. Mature thyrocytes are organized into follicular structures filled with mass colloids containing the precursor protein of thyroid hormones, thyroglobulin (TG). These follicles are the functional units of the thyroid gland [1,2,5,8]. Thyrocytes express thyrocytes-specific genes such as *TG*, thyroid peroxidase (*TPO*), and others enriched in thyrocytes, including thyroid stimulating hormone receptor (*TSHR*) and sodium/iodide symporter (*NIS*) [9,10,11,12]. Several pathologies can develop in the thyroid gland caused by thyrocyte dysfunction, such as thyroiditis, goitre, and hypothyroidism [13,14,15]. To understand the molecular mechanism of the pathophysiology of thyrocyte dysfunction, in vitro models are highly desirable. For this, immortalized thyrocyte cell lines have been established, which can retain partial thyrocyte functions such as expressing NIS and secreting TG [16,17]. However, these cells cannot form follicles; further, they are already in the committed thyrocyte lineage. While primary cultures of rodents’ thyroid glands have been developed, which appeared to retain relatively more complete physiological features of thyrocytes [18,19,20,21,22], the application of primary cultures is costly, time-consuming, and difficult to maintain consistency because of their heterogenous nature.

Embryonic stem cells (ESCs) have been utilized to study parts of the morphogenetic process of the thyrocytes, enabling researchers to examine the developmental process of the thyroid gland. ESCs can form embryoid bodies (EBs), which then develop into endodermal cells that express forkhead box protein E1 (Foxe1) and SRY-related HMG-box (SOX17). The co-expression of NKx2 home box 1 (TTF1, NKx1.2) and paired box gene 8 (Pax8) is a hallmark of thyrocyte placode formation in the ventral foregut endoderm [9,23,24,25,26]. However, this is an extremely laborious and lengthy (up to 3-weeks) procedure where consistency (i.e., clonal variation) and efficiency (i.e., the co-expression of Pax8 and TTF1) are the main challenges [24]. As such, it is not efficient to use ESC or induced pluripotent stem cells (iPSC) systems to dissect molecular mechanisms.

P19 cells are pluripotent mouse embryonal carcinoma cells derived from teratocarcinoma that can be maintained in an undifferentiated state without a feeder cell layer for an extended period. Most importantly, P19 cells exhibit all essential ESC features and can reliably differentiate into all three germ layers through first aggregating into EBs, which can then differentiate into various cell types by specific manipulations of the cell culture conditions [27]. For instance, retinoic acid (RA) induces P19 EB aggregates into neurons [28,29], whereas it induces P19 monolayer cultures into muscle cells [30].

This study aimed to first develop a method of inducing P19 stem cells into mature and functional thyroid follicle-like structures, and then to determine if this system can be adopted in molecular studies, such as gene silencing, to understand disease mechanisms. Here we report, for the first time, a reliable and efficient procedure of P19-thyrocyte differentiation to form follicular-like structures. The mature thyrocytes express key functional genes and produce and secret TG. Further, following molecular manipulation such as introduction control siRNAs, the system can still properly differentiate into mature thyrocytes, indicating the adaptability of this procedure and this new experimental system for general molecular studies of thyroid dysfunction.

## 2. Materials and Methods

### 2.1. Cell Culture and Thyrocyte Differentiation

P19 was purchased from ATCC (Cat# CRL-1825), and maintained in MEM-α (Gibco, Waltham, MA, USA, Cat# 12571-063) supplemented with 10% serum ((7.5% bovine calf serum (CS), iron fortified (ATCC, Cat# 30-2030) and 2.5% fetal bovine serum (FBS) (Atlanta Biologics, Flowery Branch, GA, USA, Cat# S11150)). For Thyrocyte differentiation, the embryoid body (EB) formation and differentiation, 2 × 10^6^ of P19 cells were grown in the medium of MEM-α with 5% Serum (75% CS + 25% FBS) plus 5% protein-free hybridoma medium (Gibco, Cat# 12040077), 1.5 × 10^−4^ M Monothioglycerol, 0.5 mM Ascorbic Acid (Sigma, Burlington, MA, USA, Cat# A4403) and 200 ug/mL Transferrin (Sigma, Cat# T8158) in a 75-T flask sitting up-right in the incubator for four days to form embryoid body. The medium was refreshed on day two. The EB were then collected by centrifugation at 100× *g* for 1 min, and transfer to a new 75-T flask with the Embryoid body endoderm differentiation (EBED) medium ((MEM-α containing 5% Serum (75% CS + 25% FBS), 5% protein-free hybridoma medium, 7.5% Knockout serum Replacement (KSR) medium (Gibco, Cat# 10828028), 10% Endoderm Induction Medium (Millipore, Burlington, MA, USA, Cat# SM302), 1.5 × 10^−4^ M Monothioglycerol, 0.5 mM Ascorbic Acid, 200 µg/mL Transferrin, 50 ng/mL human Activin A (R&D, Cat# 338-AC), 1 mIU/mL hTSH (Scripps Lab, San Diego, CA, USA, Cat# T0117) and 10 ng/mL fibroblast growth factor basic (bFGF) (Novus, Centennial, CO, USA, Cat# NBP2-35152)). The flask was set up-right in the incubator for 2 days, then refreshed the EBED medium and incubated for another two days. The EBED were collected by centrifugation at 100× *g* for 1 min, then resuspended in 5 mL of accumax (Millipore, Cat# A7089) to dissociate the EBED to single cells and plated on Matrigel coated (Corning, Corning, NY, USA, Cat# 356255) 6 cm-dish and 8-well chamber slide in the thyrocyte maturation medium ((MEM-α containing 7.5% KSR, 1 mIU/mL hTSH, 10 ug/mL Insulin (Sigma, Cat# I0516), 50 ng/mL mouse insulin-like growth factor (IGF-1; Sigma, Cat# I18779) and Potassium Iodine (KI; Sigma, Cat# 221945)). RNA was collected on Day 2, 4 and 6. On day 6, cells and conditioned medium in the 6-cm dish were collected for western, and ELISA assays, and the 8-well chambers was fixed with 4% paraformaldehyde for immunostaining. P19 differentiation experiments were repeated independently at least ten times to confirm the consistency of the process.

### 2.2. Generating Silencing RNA Control P19 Stable Cell Line

HEK-293T (ATCC CRL-3216) cell line was cultured in complete DMEM medium (Gibco, # 11,965) containing 100 U/mL penicillin, 100 mg/mL streptomycin, and 10% heat-inactivated FBS. Cells were regularly tested for mycoplasma contamination. pLKO.1 control vector plasmid were purchased from UMGC RNAi (University of Minnesota Genomics Center, RNA Interference). For lentivirus production, 2 × 10^6^ HEK-293T cells were seeded overnight in a complete DMEM medium without antibiotics in 10 cm dishes. 9.6 µg target plasmid, 7.2 µg psPAX2 packaging plasmid, and 2.4 µg pMD2.G envelope plasmid was co-transfected into cells with Lipofectamine 2000 transfection reagent (Invitrogen, Waltham, MA, USA, Cat# 11668019) following the manufacturer’s protocol. Media was changed to fresh complete DMEM medium containing 1% BSA after 6 h. Infectious lentiviruses were harvested 24 h and 48 h post-transfection and filtered through 0.45 µM pore cellulose acetate filters. Subsequently, lentiviral stocks were concentrated using the Lenti-X Concentrator (Clontech Labs, San Jose, CA, USA, Cat# 631,232) according to the manufacturer’s protocol.

For transduction, 2 × 10^5^ P19 cells were seeded in complete DMEM medium in 6 well plate overnight. Lentivirus from the pLKO.1 control vector plasmid with 8 µg/mL polybrene (Millipore TR-1003-G), were added to the P19 cells to generate P19-siRNA control. The cells were then subjected to centrifugation at 800× *g*, 37 °C for 60 min. Lentivirus was removed and changed fresh medium after 24 h, and started the puromycin selection at 48 h post transfection. Cells were selected and maintained in the same medium as described above with the addition of 1.5 µg/mL puromycin. For single cell clone isolation, 10 cells were seeded in a 10-cm dish. Following one week, colonies reached a size visible to the naked eye. These colonies were then picked using 20 µL of trypsin and transferred into 96-well plate.

### 2.3. Primary Follicular Thyrocytes Culture

This procedure was adapted from a previous study isolating follicles from rodents [19]. All experimental procedures were conducted according to NIH guidelines, and protocols were approved by University of Minnesota Institutional Animal Care and Use Committee (IACUC) (2203-39874A, approved 2 June 2022), and Institutional Biosafety Committee (IBC) (2012-38742H, approved 2 February 2022). Wild-type C57BL mice were obtained from Jackson Laboratory and housed in the University of Minnesota animal facilities. Two male mice (3–4-month-old, 35–40 g) were euthanized by CO_2_. The trachea, where the thyroid gland is attached to, was excised and placed in a sterile plate containing DMEM/F12 (1:1) (Gibco, Cat# 11330-032), under light microscope the lobes of thyroid gland were isolated from the trachea and then digested for 45 min at 37 °C, agitated briefly every 5 min. The digestive media contained 1 mg/mL of collagenase type II (Worthington-biochem, Lakewood, NJ, USA, Cat# LS004177), 0.4 mg/mL of trypsin inhibitor (Sigma-Aldrich, Burlington, MA, USA, Cat# T6522), and 3 mg/mL of Dispase II (Sigma-Aldrich, Cat# D4693) dissolved in DMEM/F12. All the following steps were carried out in a sterile environment, in a laminar hood. The digestive media was replaced by new media of DMEM/F12 supplemented with 10% Nu-serum (Corning, Cat# 355500). The pellet was dissociated by triturate with 1000 µL pipette and passed through 200 µm filter to remove debris and undissociated parts of the thyroid gland. This process was repeated until all follicles were dissociated and no more clumps were detected. Then, the suspension that contained single cells and follicles was passed through 50 µm filter; the follicles were retained on the filter while all single cells passed the filter, then to collect the follicles, the 50 µm filter was washed several times with the same media. The collected follicles were centrifuged at 400× *g* for five minutes, then resuspended the pellet with 200 µL of the same media. Six wells plate was used to cultivate the primary follicular thyrocytes, coated with collagen type I (Corning, Cat# 354236). The collected follicles were seeded in droplet fashion and incubated in 5% CO_2_, 37 °C for 30 min in the incubator. The maintained media was added, which consisted of DMEM/F12 supplemented with 10% Nu-serum, 2 ng/mL glycl-L-histidyl-L-lysine (Sigma, Cat# M6145), 3.5 ng/mL hydrocortisone (Sigma, Cat# H0888), 10 nM potassium iodide, 0.5µM ascorbic acid (Sigma, Cat# A4403), 10 µg/mL human transferrin (Sigma, Cat# T8158), 10 µg/mL bovine insulin (Sigma, Cat# I0516), 10 ng/mL somatostatin (Sigma, Cat# S1763), and 1 mIU/mL TSH. The media changed every other day, and the primary follicular thyrocytes culture retained its physiological function for up to 14 days. Primary follicular thyrocytes were repeated independently at least five times to confirm the consistency of the process.

### 2.4. Quantitative Real-Time PCR

Cells were homogenized in trizol (Thermo Fisher, Waltham, MA, USA, Cat# 15596026) by using cell scraper to extract mRNA, and 2 mg of total RNA was reverse transcribed to cDNA with a reverse transcript kit (Thermo Fisher, Cat# 4368819). Quantitative real-time PCR (qPCR) was performed using the SYBR™ Green PCR Master Mix (Thermo Fisher, # K0253) on a real-time PCR machine (QuantStudio™ 3 Real-Time PCR System, Applied Biosystems, Waltham, MA, USA). The real-time PCR reaction was performed in duplicate for each sample. All primers are described in Table 1.

### 2.5. Immunostaining

8-well chamber slides consisting of either P19-differentiated thyrocytes or primary follicular thyrocytes were fixed with 4% paraformaldehyde (Sigma, Cat# P6148) for 4 h at room temperature and stored in PBS at 4 °C. 5% normal donkey serum (Sigma, Cat# D9663) and 0.1% Triton-X in PBS used to block the cells for 1 h at room temperature. Thyroglobulin (TG) detected by rabbit monoclonal TG antibodies (Abcam, Cat# ab156008) and Sodium/iodide symporter (NIS) detected by rabbit polyclonal NIS antibodies (Proteintech, Rosemont, IL, USA, Cat# 24324), were incubated overnight at 4 °C in the dark chamber, then labelled by secondary antibody Alexa-488-conjugated donkey antirabbit (Jackson immune, Philadelphia, PA, USA, Cat# 711-546-152) or Cy5-conjugated donkey antirabbit (Jackson immune, Cat# 711-175-152) for 2 h at room temperature. Slides were stained with DAPI for 10 min at room temperature and mounted with antifading media Vestasheild (Vector lab, Newark, CA, USA, Cat# H-1000). All images were acquired by using Olympus fluoview BX2 upright confocal microscope. Image J (version 1.53) was utilized to create heatmaps and histograms figures.

### 2.6. ELISA

Culture medium was collected after two days of stimulation. We use TSH stimulation in P19 differentiated thyrocytes by adding tenfold of TSH concentration compared to the basal physiological levels (1 mIU/mL). In contrast, primary follicular thyrocytes were depleted from TSH and stimulated with basal TSH concentration (1 mIU/mL). The presented data were P19 differentiated thyrocyte conditioned medium, primary follicular thyrocyte conditioned medium, and controlled culture medium of each type of culture. This was followed by cleaning up the conditioned medium from debris and dead cells by centrifugation at 400× *g* for 5 min at room temperature, then 2 mL of each medium were concentrated using ultra centrifugal filter 50 kDa (Sigma, Cat# UFC9050). 20 µL of each concentrated samples in triplicate were used to analyse the TG concentration using the mouse TG ELISA kit following the manufactural protocol (Novus Biologicals, Centennial, CO, USA, Cat# NBP3-08183).

### 2.7. Statistical Analysis 

The data followed a normal distribution, as assessed by Shapiro–Wilk test presented as mean ± SE. For Figures 2, 3 and 6, one-way ANOVA followed by Bonferroni post hoc test was performed. For Figure 7A, two-way ANOVA followed by the Bonferroni post hoc test was performed. The comparison was considered statistically significant when *p* values < 0.05, and SPSS was utilized for statistical analysis. 

## 3. Results

### 3.1. Generation and Characterization of P19 Differentiated Thyrocyte

The established procedure to induce P19 differentiation into thyrocytes is shown in Figure 1. Firstly, P19 cells were seeded in a non-adhesive flask to form embryoid bodies (EBs) for four days. Activin A, TSH, bFGF, and 10% of endoderm induction medium were added to induce EB endoderm formation and differentiation (EBED) containing the thyrocyte progenitor lineage for four more days. The EBED aggregates were then dissociated into single cells and plated on the Matrigel-coated dish in the serum-free medium supplemented with TSH, Insulin, IGF-1, and KI, allowing the cells to differentiate into thyrocytes and form follicle-like structures. Mature thyrocytes were polarized with their apical membrane facing the colloid, the inner part of the follicle, and the basal membrane facing the outer part of the follicle, as shown in the bottom right of Figure 1. In these structures, thyrocytes are functional, as indicated by their ability to produce and release TG.

P19 cells transitioning through each stage were monitored by examining the expression of key stage-specific genes. The stemness properties decreased when P19 undifferentiated cells transitioned to EB and EBED stages. As shown in (Figure 2A), the stemness gene expression of Octamer-binding transcription factor (*OCT4)* and Reduced expression1 (*REX1*) decreased both in EB and EBED compared to P19 undifferentiated cells. Adding activin A and other growth factors induced EBs to form EB endoderm. This induction was confirmed by examining the endoderm gene expression of *α-fetoprotein* and GATA binding protein 4 (*GATA4*) that is increased in EBED after four days of induction (Figure 2B). The gene expression of *Foxa2* and *SOX17* promotes the EBED commitment to definitive EBED containing thyrocyte progenitors. P19 EBED exhibited an increase in gene expression of *Foxa2* and *SOX17* compared to P19 undifferentiated cells, as shown in (Figure 2C). Seeding the definitive EBED containing thyrocyte progenitors in Matrigel-matrix induces thyrocyte maturation, and the mature thyrocytes can be maintained for up to 6 days. Thyrocytes maturation was monitored by evaluating the expression of key functional genes such as *TG*, *TPO*, *NIS* and *TSHR*. In Figure 3A, *TG* and *TSHR* gene expression increased gradually until day six. The gradual increase of *TG* indicates that as the thyrocytes grow and form follicles (see next section), thyrocytes will produce and secret TG continuously. *TPO* and *NIS* are required for thyrocytes maturing functioning, but they are not involved in growing thyrocytes. In P19 differentiated thyrocytes, the gene expression of *TPO* and *NIS* increased on day four and then decreased. We examine several transcription factors identified in thyrocyte development to determine the definitive EBED commitment and destined to differentiate into thyrocyte. Gene expression increased of *Pax8*, *TTF1*, *TTF2* and *Hhex* during thyrocyte maturation (Figure 3B). The overlapping of these transcription factors validates the differentiation of P19 into thyrocytes.

### 3.2. P19 Differentiated Thyrocytes Form Follicle-like Structures Resembling Primary Follicles

Functional thyrocytes produce and secret TG to the colloid located in the follicle’s center. Follicles form through thyrocytes rearrangement so that the apical surface of polarized thyrocyte faces the center, the colloid. As shown in Figure 4A top panel (middle and right), the expression and accumulation of intracellular TG in P19 differentiated thyrocytes were comparable to that of primary follicular thyrocyte cultures. TG was distributed evenly throughout thyrocytes in both primary follicular thyrocytes and P19 differentiated thyrocytes (magenta arrows in Figure 4A top panel middle and right). In contrast, P19 stem cells did not express TG (Figure 4A top panel left). In the thyroid gland, thyrocytes form follicles by polarizing the apical surface toward the colloid where TG is secreted (the sphere’s center). P19 differentiated thyrocytes formed follicle-like structures, and appeared in a hemispherical or circular shape because the coating material of the dish did not provide a 3D environment, which remains to be improved (see Section 4). Both P19 differentiated thyrocytes, and primary follicular thyrocytes exhibited very similar circular shapes (Figure 4A top panel middle and right). In addition, both cultures secreted TG, which accumulated in the center of follicle-like structures (Figure 4A bottom panels). Despite the continuously increased expression of TG in the P19 differentiated thyrocytes for up to day 6 of maturation, TG protein expression was lower in P19 differentiated thyrocytes as compared to TG expression in primary follicular thyrocytes extracted from follicles of 3-month-old thyroid glands. This suggests a room for improvement to optimize the functional maturation of this in vitro thyrocyte differentiation and maturation model.

To further demonstrate the follicle-like structures formed by P19 differentiated thyrocytes, we assessed NIS expression by immunostaining. As expected, the polarized thyrocytes in the follicles expressed and distributed NIS mainly in the basal membrane facing the outer surface of the follicle, different from intracellular TG distributed evenly thyrocytes. Follicles formed from primary follicular thyrocytes exhibited a very similar NIS expression pattern as compared to that of P19 differentiated thyrocytes (Figure 4B lower panel). Similarly, the NIS (red stain) distribution pattern in the follicular-like structures showed NIS localization in one side of the nuclei (DAPI), again indicating polarization of these thyrocytes (white arrows in Figure 4B middle and left upper panels).

In summary, these in vitro differentiated thyrocytes formed unique structures where only the basal surface of these differentiated cells expressed NIS, supporting that these differentiated cells indeed formed follicle-like structure (Figure 4B lower panel). For a control, P19 undifferentiated cells did not show this specific NIS expression pattern, nor did they form follicle-like structures (Figure 4B left panels).

### 3.3. P19 Differentiated Thyrocytes Respond to TSH Stimulation

The most important physiological property of thyrocytes is their response to TSH stimulation [16,31] by increasing TG secretion [32]. We monitored P19 differentiated thyrocytes and primary follicular thyrocyte cultures to determine their abilities to increase TG secretion into the culture medium in response to TSH stimulation for two days (indicated as a conditioned medium). TG secreted into the medium was determined using ELISA. Culture medium was used as a control. Since P19-thyrocyte culture readily contained a basal (physiological, 1 mIU/mL) level of TSH as required for proper differentiation, a higher dose of TSH (tenfold of basal level (10 mIU/mL) was used to stimulate TG secretion from P19-thyrocytes, whereas primary follicular thyrocytes were stimulated with the basal physiological concentration of TSH (1 mIU/mL). As shown in Figure 5, P19 differentiated thyrocytes responded to TSH stimulation by increasing TG secretion for ~30%. Primary follicular thyrocytes responded to TSH stimulation by increasing TG secretion for ~40%. This result clearly demonstrated that P19-differentiated thyrocytes indeed can respond to TSH stimulation by increasing TG secretion, a critical physiological feature of functional thyrocytes.

### 3.4. P19-siRNA Control Can Differentiate into Mature Thyrocytes

We next determined if this P19-thyrocyte differentiation system and procedure would be appropriate for general molecular/genetic studies, i.e., if the system could tolerate molecular/genetic manipulation procedure such as gene silencing without affecting its thyrocyte differentiation potential. We first established P19 clones containing a siRNA control vector, name P19-siRNA control. The P19-siRNA control was then subjected to the established thyrocyte differentiation procedure, and was monitored for the expression of key markers during thyrocyte differentiation. As shown in Figure 6A, P19-siRNA control clone retained a differentiation potential like the wild type P19 in that their stemness genes such as *OCT4* and *REX1* were all gradually decreased in EB stages (Figure 6A). Furthermore, P19-siRNA control generated EBED, confirmed by the increase of *alpha-fetoprotein* and *GATA4* gene expression (Figure 6B). Finally, the increase in *Foxa2* and *SOX17* gene expression in P19-siRNA control also exhibited a similar trend as compared to wild type P19, confirming that EBED indeed contained thyrocyte progenitors (Figure 6C). These results showed that the differentiation potential of P19-siRNA control, using this established procedure, was not affected by the transfection procedure or the introduction of control siRNA.

We next evaluated P19-siRNA control-differentiated thyrocytes in terms of gene expression specific to thyrocyte function. As shown in Figure 7A, the expression of mature thyrocyte-specific genes such as *TG*, *TPO* and *TSHR* were comparable between P19 wild type- and P19-siRNA control-differentiated thyrocytes. *NIS* is also known to be enriched in thyrocytes, and both P19-wild type and P19-RNA control displayed a similar pattern of *NIS* gene expression.

## 4. Discussion

Because of its ability to maintain embryonic stem cell features and since it requires no feeder layers, the P19 pluripotent embryonal carcinoma cell line has been widely used to differentiate into various cell types for molecular studies. This current study is the first to develop a reliable and efficient differentiation procedure to induce P19 cells to form functional thyrocytes.

Several factors are known to be involved in inducing EBED to destined and committed thyrocyte progenitor lineage. Transcription factor TTF1 (NKX2-1) is the first identified transcriptional factor that regulates *TG* promoter, considered to be most crucial for thyrocytes progenitors’ commitment. It also regulates the expression of other genes, such as *TPO*, *TSHR* and *NIS* [33,34,35]. Pax8 is another master transcriptional factor because it is required to activate the transcription of *TG*, *TPO* and *NIS* in thyrocytes [33,36,37]. Additionally, Pax8 synergizes with TTF1 to stimulate transcriptional activation of *TG* gene [38] and other thyroid enriched genes including dual oxiadase-2 (*Doux-2*), and deiodinases-1 (*Dio1*) [39,40]. Along with Pax8 and TTF1, other factors, such as Hhex and Foxe1, were identified as “thyroid-specific gene expression program” responsible for thyrocyte differentiation [24], however, it was unclear whether those factors have to be co-express in the cells simultaneously or, consequently manner.

In order to study thyrocyte development, mouse and human ESCs were first used to differentiate into thyrocytes [9,23,24,25]. These earlier studies that aimed to differentiate ECS cells into thyrocytes revealed that either Pax8 or TTF1 was expressed; however, expressing a single transcription factor through transfection failed to induce ESC cells differentiation into thyrocytes [24,25]. Therefore, ESC has to be transfected with both Pax8 and TTF1 to induce thyrocyte differentiation [24,41] and mature thyrocytes that express TG and form follicles where TG accumulates in the colloid [24,25,41]. Importantly, our established P19-thyrocyte differentiation model recapitulates the entire thyrocyte differentiation, maturation, and functional maintenance process without the need for transfecting the cultures with exogenous factors.

Taking into consideration ESC’s clonal variability and its inconsistent transfection efficiency (such as to co-express multiple transcription factors), using ESC cells as in vitro models for studying thyrocyte development is difficult. The capacity of P19 to differentiate into EBED and then commit to differentiation into functional thyrocytes without introducing exogenous factors is useful. Furthermore, P19 stem cell cultures require no feeder layers and can be maintained in their stem cell state for extended passages. These features make the P19-thyrocyte differentiation system an attractive in vitro experimental model for understanding molecular players and pathways underlying the formation and maintenance of healthy and functional thyrocytes.

The thyroid gland contains various cell types, including thyrocytes and parafollicular cells such as C-cell, which could also complicate experimental results. Most studies have utilized primary cultures as in vitro models to examine their molecular mechanism of thyroid gland pathology. For this, the thyroid gland has to be dissociated into single cells for culture; however, during this process, thyrocytes lose their orientation and physiological function such that they are not able to synthesize or secret TG [21]. An alternative approach is to partially dissociate the thyroid gland to reserve follicles that resemble follicular units of the thyroid gland and can be used to study molecular mechanisms of pathophysiology [2,5,19]. However, primary thyroid cultures are inherently inconsistent and inefficient because of both technical and biological reasons (such as individual variability and the presence of other cells like C-cells). A most critical limitation in using primary thyroid cultures is the lack of differentiation and maturation processes in this type of culture. In contrast, P19 differentiated thyrocytes culture system recapitulates the entire thyrocyte differentiation and maturation process and is uniquely suitable for studying thyrocyte development and functional maintenance in vitro for a reasonable duration (up to 6 days). With specific genetic manipulations, this P19-thyrocyte model provides a reliable, and cost-and labour-efficient system for detailed and systemic molecular studies of events maintaining healthy thyrocytes, as well as pathological events leading to their dysfunction.

In this current model, P19 differentiated thyrocytes are seeded in Matrigel-matrix and closely resemble primary follicular thyrocytes in terms of function and the 2D structure and morphology (follicle-like structures). A further future improvement may be desirable, such as to exploit specialized Matrigel-matrix in seeding thyrocyte progenitors to facilitate the formation of a 3D structure. Nevertheless, our current model readily recapitulates the functionality of primary follicular thyrocytes, enabling systemic molecular studies where reproducibility and efficiency are most crucial.

Finally, to apply this system in a more general term, we examined whether this thyrocyte differentiation procedure coupled with the P19 model system can tolerate the common gene silencing procedure (Figure 6 and Figure 7, Appendix A).

All these data confirm that this newly developed P19-thyrocyte differentiation system can tolerate molecular/genetic manipulation procedures (such as introducing siRNA for studying specific gene function) and retain their capacity for proper differentiation into mature and functional thyrocytes. This provides a powerful tool and experimental system allowing systemic in vitro studies to investigate molecular mechanisms underlying healthy thyrocytes development and function, as well as those leading to pathological processes of various thyroid diseases. This is a much more feasible, cost-effective and time-efficient model than animal models or models derived from primary cultures. Nevertheless, there are rooms to improve, such as the efficiency of P19-differentiation and the formation of follicles using 3D culture.

As introduced earlier, this study was prompted by our surprising finding of mature thyrocyte malfunction caused by the disruption of *Crabp1* gene, because CKO mice developed adult-onset primary hypothyroidism with apparent defected thyrocytes, but the mechanism remains unknown [4]. This newly developed P19-thyrocyte model will permit further systemic studies to be conducted to determine the detailed mechanisms underlying the pathophysiology of thyroid diseases, including hypothyroidism of CKO mice.

## 5. Conclusions

This work reported for the first time the development of a reliable and cost- and labour-efficient in vitro model of thyrocyte differentiation, maturation, function and forming follicular-structures using P19 stem cell cultures. The P19-thyrocyte differentiation/maturation process spans several transiting phases including EBED differentiating to thyrocyte progenitors and then into functional thyrocytes to form follicle-like structures. The P19-thyrocyte-formed follicle-like structures can secrete TG into the culture medium and respond to TSH stimulation, further supporting their capacity for the expected physiological function. This P19-thyrocyte differentiation model system tolerates molecular/genetic manipulation, enabling efficient and cost-effective studies of molecular mechanisms underlying the pathophysiology of thyroid diseases. This also provides a potential tool applicable in regenerative medicine. 

## Figures and Tables

**Figure 1 cells-13-01844-f001:**
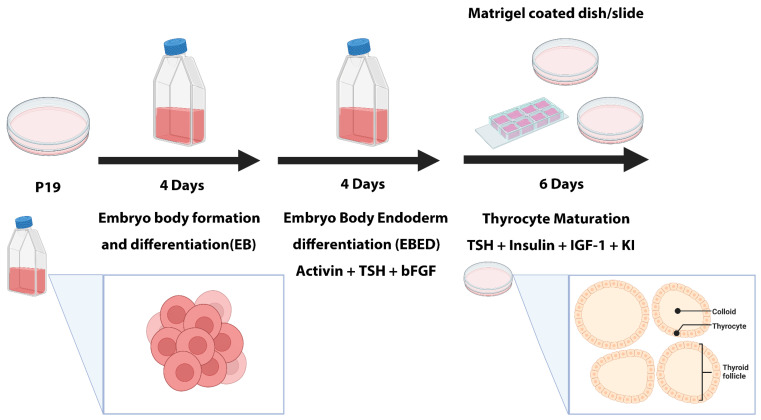
Schematic procedure of P19 differentiation into thyrocytes. Undifferentiated P19 cells are seeded for four days to form embryoid bodies (EBs), followed by four days of incubation with a supplemented medium to differentiate into EB endoderm differentiation (EBED) containing thyrocyte progenitors. The aggregates of EBED are dissociated into single cells and seeded on a Matrigel-coated dish, where cells matured to form follicle-like structures as shown in the bottom of this figure.

**Figure 2 cells-13-01844-f002:**
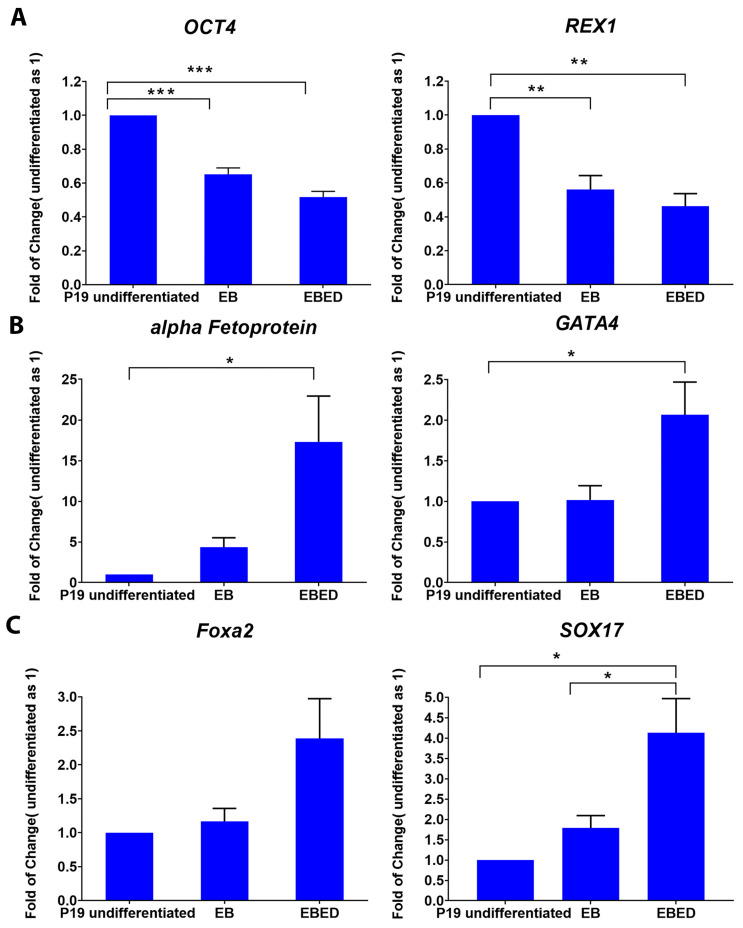
Gene expression patterns at different stages of P19 transition from the undifferentiated state to EB endoderm differentiation (EBED) stage. (**A**) Expression of stemness genes: *OCT4* and *REX1* in undifferentiated P19 cells, EBs, and EBED. (**B**) Expression of endoderm-specific genes: *α-fetoprotein* and *GATA4* in undifferentiated P19 cells, EB, and EBED. (**C**) Expression of genes specific to thyrocyte progenitors and definitive EBED: *Foxa2* and *SOX17* in undifferentiated P19 cells, EB, and EBED. The data show means ± SE of 3 independent experiments. One-way ANOVA followed by Bon-ferroni post hoc analysis was performed, * *p* <0.05, ** *p* < 0.01, *** *p* < 0.001.

**Figure 3 cells-13-01844-f003:**
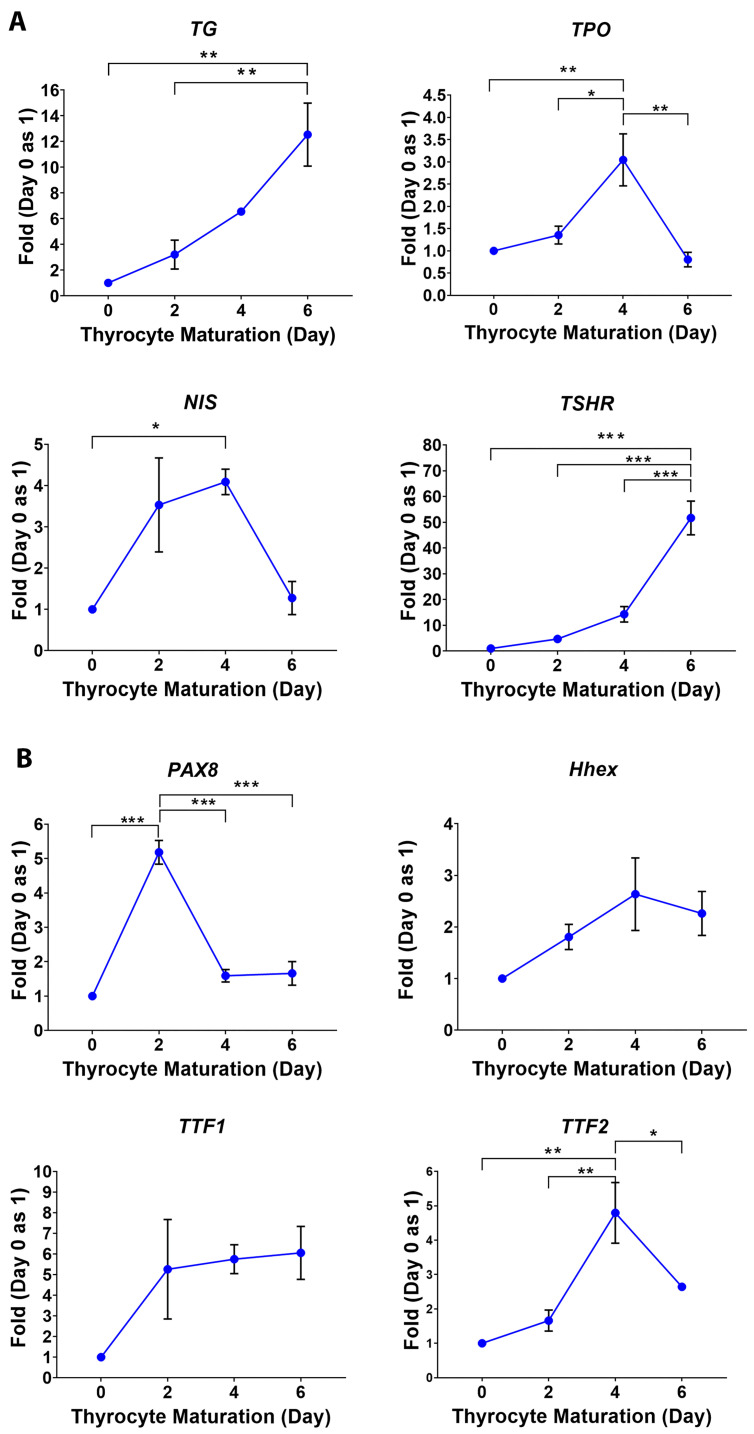
Gene expression patterns of P19 differentiated thyrocytes during maturation. (**A**) Expression of genes specific to thyrocyte functions: *TG* and *TPO*, and those enriched in thyrocytes *NIS* and *TSHR*. (**B**) Expression of transcription factors: *Pax8*, *TTF1*, *TTF2* and *Hhex*. The data show means ± SE of 3 independent experiments. One-way ANOVA followed by Bon-ferroni post hoc analysis was performed, * *p* <0.05, ** *p* < 0.01, *** *p* < 0.001.

**Figure 4 cells-13-01844-f004:**
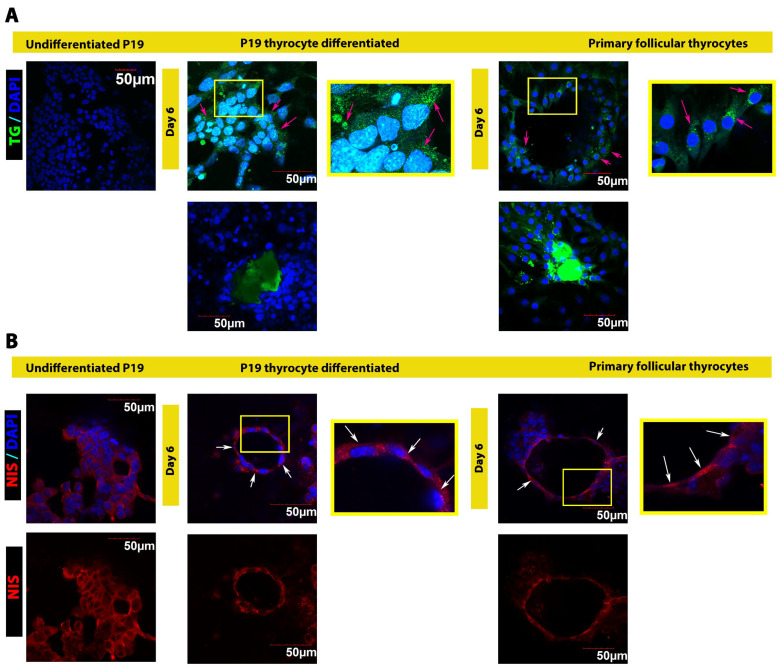
Immunostaining of TG and NIS protein in undifferentiated P19, P19 differentiated thyrocytes and primary follicular thyrocytes. (**A**) Upper panel: intracellular TG protein expression (TG, green stain) in undifferentiated P19, P19 differentiated thyrocytes and primary follicular thyrocytes. The intracellular TG was distributed evenly throughout thyrocytes, indicated by magenta arrows. Lower panel: TG staining in the colloid of follicle-like-structures formed by P19 differentiated thyrocytes and primary follicular thyrocytes. The heatmaps/histograms were obtained, shown in Appendix A. (**B**) Upper panel: NIS expression (NIS, red stain) in undifferentiated P19, P19 differentiated thyrocytes, and primary follicular thyrocytes. NIS was detected mainly on one side (red stain) of the cells (nuclei stained with DAPI), indicating the polarization of thyrocytes (white arrows). The lower panel showed the polarization of NIS pattern.

**Figure 5 cells-13-01844-f005:**
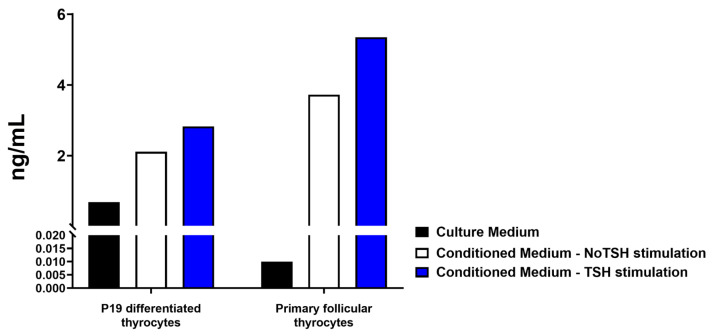
TSH stimulates TG secretion. TG secreted from P19-differentiated thyrocytes and primary follicular thyrocytes was detected by ELISA. Condition media were collected from three sets of experiments with two days of TSH stimulation. Culture medium was included as the control in each experiment. Primary follicular thyrocyte culture was stimulated with a physiological dose of TSH (1 mIU/mL); whereas P19-thyrocyte culture readily contained a basal level TSH (1 mIU/mL), therefore a higher dose of TSH (10 mIU/mL) was used to stimulate TG secretion.

**Figure 6 cells-13-01844-f006:**
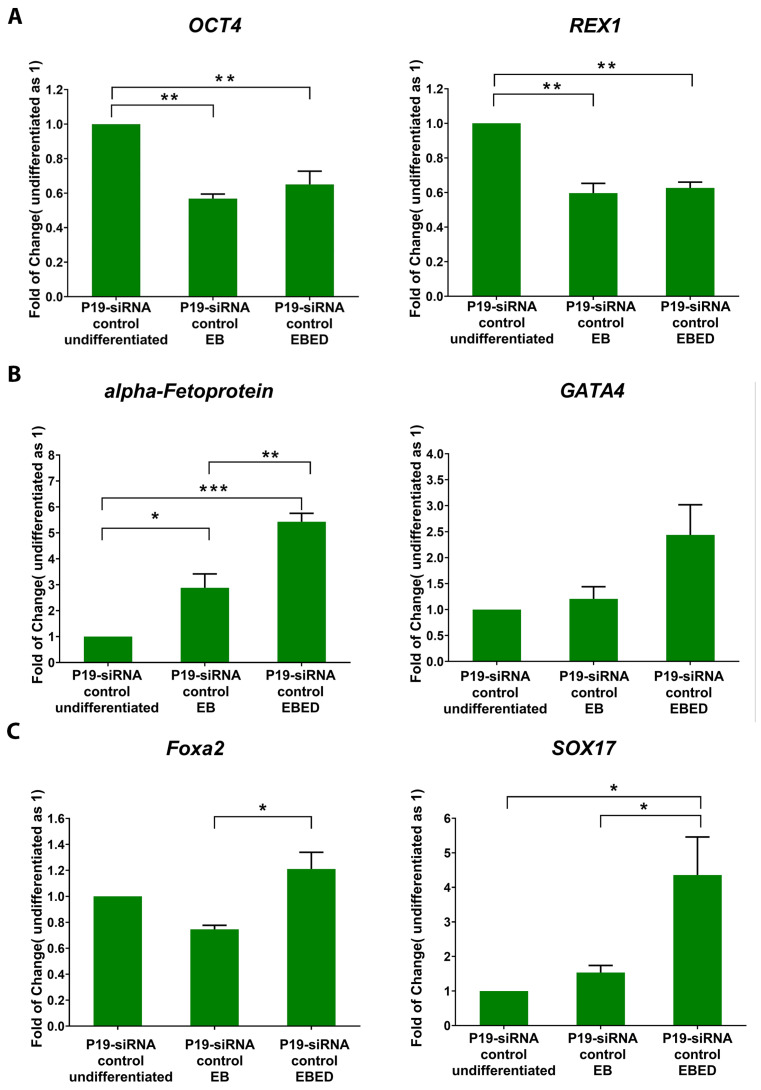
Gene expression patterns at different stages of P19-siRNA control culture transitioning from undifferentiated state to EB endoderm differentiation (EBED) stage. (**A**) Expression of stemness genes *OCT4* and *REX1* in P19-siRNA control, including stages of undifferentiated, EBs, and EBED. (**B**) Expression of endoderm-specific genes: *α-fetoprotein* and *GATA4* in P19-siRNA control, including stages of undifferentiated, EB, and EBED. (**C**) Expression of genes specific to thyrocyte progenitors and definitive EBED: *Foxa2* and *SOX17* in P19-siRNA control, including undifferentiated, EB, and EBED. The data show means ± SE of 3 independent experiments. One-way ANOVA followed by Bonferroni post hoc analysis was performed, * *p* <0.05, ** *p* < 0.01, *** *p* < 0.001.

**Figure 7 cells-13-01844-f007:**
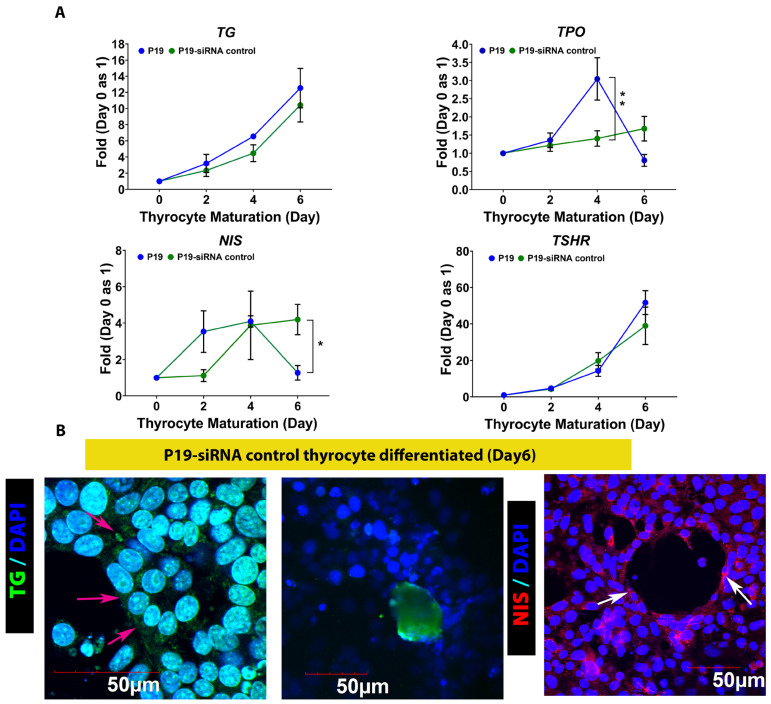
Gene expression patterns and immunostaining of P19-siRNA control differentiated into thyrocytes. (**A**) Expression patterns of genes specific to thyrocyte functions: *TG* and *TPO*, and those enriched in thyrocytes, *NIS* and *TSHR* in P19-wild type and P19-siRNA control cultures thyrocytes differentiated. (**B**) Immunostaining of TG and NIS in P19-siRNA control cells-differentiated follicle-like structures. Left panel, intracellular TG was distributed evenly in P19-siRNA control cells-differentiated thyrocytes (magenta arrows), and secreted TG accumulated inside the follicle (middle panel). Right panel, NIS was detected in P19-siRNA control cells-differentiated thyrocytes only on one side of the cells, indicating the expected polarity of these differentiated thyrocytes (white arrows). The heatmaps/histograms of TG and nuclear staining were obtained, shown in Appendix A. Panel A data show means ± SE of 3 independent experiments. Two-way ANOVA followed by Bonferroni *post hoc* analysis was performed, * *p* <0.05, ** *p* < 0.01.

**Table 1 cells-13-01844-t001:** Primers sequence for qPCR.

Gene Name	Primer Sequence
*OCT-4*	Forward: 5′-CAGCAGATCACTCACATCGCCA-3′; Reverse: 5′-GCCTCATACTCTTCTCGTTGGG-3′
*REX1*	Forward: 5′- GAGAAGAGGAGGATTGCTCACG-3′; Reverse: 5′-CAATGTGCGTGTCTTCAGTGGC-3′
*a-Fetoprotein*	Forward: 5′-GCTCACATCCACGAGGAGTGTT-3′; Reverse: 5′-CAGAAGCCTAGTTGGATCATGGG-3′
*GATA-4*	Forward: 5′-GCCTCTATCACAAGATGAACGGC-3′; Reverse: 5′-TACAGGCTCACCCTCGGCATTA-3′
*Foxa2*	Forward: 5′-CGAGCACCATTACGCCTTCAAC-3′; Reverse: 5′-AGTGCATGACCTGTTCGTAGGC-3′
*SOX17*	Forward: 5′-GCCGATGAACGCCTTTATGGTG-3′; Reverse: 5′-TCTCTGCCAAGGTCAACGCCTT-3′
*TSHR*	Forward: 5′-GCTGTCGTTGAGTTTCCTCCAC-3′; Reverse: 5′-CTGCTCTCATTACACATCAAAGAC-3′
*TPO*	Forward: 5′-GAGAGGCTCTTCGTGCTGTCTA-3′; Reverse: 5′-AGGCGTGACAAGCCACAGAACT-3′
*TG*	Forward: 5′-TTGTAGCCTGGAGAGTCAGCAC-3′; Reverse: 5′-CACTGCACATCTTTCCTGGTGG-3′
*NIS*	Forward: 5′-CATGCCATTGCTCGTGTTGGAC-3′; Reverse: 5′-GCCATAGCGTTGATACTGGTGG-3′
*TTF1*	Forward: 5′-CAGGACACCATGCGGAACAGC-3′; Reverse: 5′-GCCATGTTCTTGCTCACGTCCC-3′
*Pax8*	Forward: 5′-TGCTCAGCCTGGCAATGACAAC-3′; Reverse: 5′-ACGAAGGTGCTTTCGAGGACCA-3′
*TTF2*	Forward: 5′-AACAGCATCCGCCACAACCTCA-3′; Reverse: 5′-AGGAAGCTGCCGCTTTCGAACA-3′
*Hhex*	Forward: 5′-CGGTCAAGTGAGGTTCTCCAAC-3′; Reverse: 5′-CTCGGCGATTCTGAAACCAGGT-3′

## Data Availability

Data are contained within the article.

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
