# Peer review of "Modelling Functional Thyroid Follicular Structures Using P19 Embryonal Carcinoma Cells"

_cells, 2024, doi:10.3390/cells13221844_

Round 1

Reviewer 1 Report

Comments and Suggestions for Authors

I would like to thank Najjar et al for this interesting paper on creating functional thyroid follicular structures from P19 stem cells. It is a well presented essay and quite original since it describes for the first time such a procedure. I have only minimal comments as follows:

Introdution should be shortened at least by a paragraph. 

Line 61. The "largest" or the "main" concern

Line 150. You mention Minnesota twice, please rephrase

Line 161. "took place" or "were carried out"

Line 174. lysine (Sigma)

Line 183. "reverse transribed"

Lines 456-465: intensely, extremely, extremely: please avoid using such intense words

Line 467: thyroid tissue per se does not contain parathyroid cells. The parathyroid glands are resting on the thyroid tissue but the two types of glands are totally separated. Please correct

Line 470. lose "their"

Line 501. mechanism "remains" 

Comments on the Quality of English Language

Minor corrections as described above

Author Response

Reviewer  1

Introduction should be shortened at least by a paragraph.

      Reply – Introduction was shortened by deleting parts of 2nd and 3rd paragraphs.

Line 61. The "largest" or the "main" concern

      Reply – Text spanning this sentence was deleted in order to shorten “Introduction” (see comment 1).

Line 150. You mention Minnesota twice, please rephrase

      Reply – This was corrected as “University of Minnesota” line 146.

Line 161. "took place" or "were carried out"

      Reply – As suggested, this was rephrased as “were carried out” line 157.

Line 174. lysine (Sigma)

      Reply – This was corrected by inserting a space, line 170.

Line 183. "reverse transcribed"

      Reply – The wording was corrected, as “reverse”, line 179.

Lines 456-465: intensely, extremely, extremely: please avoid using such intense words

      Reply – These words were deleted, in 447-455.

Line 467: thyroid tissue per se does not contain parathyroid cells. The parathyroid glands are resting on the thyroid tissue but the two types of glands are totally separated. Please correct

      Reply - Thank you for correction. “Parathyroid cells” were deleted from this section of the text.

Line 470. lose "their"

      Reply – “Its” was corrected as “their” line 460.

Line 501. mechanism "remains" 

      Reply – “Remain” was corrected as “remains” line 498.

Reviewer 2 Report

Comments and Suggestions for Authors

The authors attempted to develop an in vitro thyrocyte model using P19 embryonal carcinoma cells. The authors showed that thyrocyte-like cells can be induced by stimulating P19 cells with TSH and other substances. However, the reproducibility of these experiments is unclear, and research should be conducted with sufficient planning. My concerns are as follows.

#1. This title has not been sufficiently demonstrated in this study. Although the title and Figure 1 state that thyroid follicular structures are formed from P19 cells, there are no pictures to prove this, such as those taken with a phase-contrast microscope or pictures of serial sections of paraffin-embedded tissue samples. The title states that “P19 stem cells” are used in the study, but this should be “P19 embryonal carcinoma cells”, which is misleading to the readers.

#2. The experiments in Figures 2, 3, 5, 6, and 7 must be statistically analyzed using data from at least three or more treatments or trials. The values shown in the figures are not stated as mean values, and standard deviations and standard errors are not stated. Statistical analyses, including multiple comparisons, were not performed, and it is difficult to say that the data are reliable.

#3. In Figures 4 and 7, immunofluorescence staining (IF) of thyroglobulin (TG) and DAPI is shown, but it seems that the gain is set high in the P19 cells as compared with that in primary thyrocytes, and the fluorescence is close to saturation. It should be imaged under appropriate conditions. In addition, the fluorescence intensity of IF varies greatly depending on the setting; therefore, it is not quantitative. If the authors want to compare the amounts in the histological specimens, they should use immunohistochemistry with DAB.

#4. As shown in Figure 5, P19 cells differentiated into thyrocytes secreted less thyroglobulin in response to TSH than primary cultured thyrocytes. Thyroglobulin production is a fundamental function of the thyroid follicular cells. Low response to TSH stimulation is a major problem when using P19 cells as a model of thyroid follicular cells.

#5. The experiments in Figures 6 and 7 show that the negative control siRNA served as the negative control. This negative control siRNA has been widely used in other studies and has almost no value. It is necessary to compare its function with that of siRNAs for certain thyroid-related genes such as TTF1 and PAX8.

#6. “Embryo body” should be “embryoid body.”

Comments on the Quality of English Language

Moderate editing of the English language is required.

Author Response

Reviewer (2)

#1. This title has not been sufficiently demonstrated in this study. Although the title and Figure 1 state that thyroid follicular structures are formed from P19 cells, there are no pictures to prove this, such as those taken with a phase-contrast microscope or pictures of serial sections of paraffin-embedded tissue samples. The title states that “P19 stem cells” are used in the study, but this should be “P19 embryonal carcinoma cells”, which is misleading to the readers.

Reply - Thank you. The title was corrected as “….P19 embryonal carcinoma…. 

About follicular structure, thank you for calling attention to the structure of the follicles in Figure 1. All the images that were obtained for P19-differentiated thyrocytes and primary follicular thyrocyte cultures showed a hemisphere shape where thyroglobulin was accumulated in the center (colloid), as shown in Figure 4.A (lower panels, middle and right). Thyrocytes do not form follicles unless they are polarized where sodium iodide symporter (NIS) is expressed only on the basal surface of the polarized cell. To confirm that the hemispheres in these cultures represent parts of the follicles, we utilized NIS marker to show polarization of these cells (Figure 4.B, yellow-highlighted, middle and right squares).

The goal of this current study is to establish a reliable and efficient procedure to differentiate P19 into functional thyrocytes that can advance to polarization in order to form follicle-like structures. Our next step will advance to the Matrigel system that allows 3D cultures to form. This was added in Discussion, lines 492-4.

#2. The experiments in Figures 2, 3, 5, 6, and 7 must be statistically analyzed using data from at least three or more treatments or trials. The values shown in the figures are not stated as mean values, and standard deviations and standard errors are not stated. Statistical analyses, including multiple comparisons, were not performed, and it is difficult to say that the data are reliable.

      Reply – We appreciate reviewer’s comments and recognize that statistical analysis of biomedical experiments is important in most cases. However, it is also true that most studies describing stem cell differentiation experiments have provided either data of immunostaining of a specific protein (marker) that is specifically expressed in a cell type such as neuron, muscle, etc., or data illustrating the functionality of the specific cell type, such as mechanical stimulation (response), secreting hormones (action),…etc. These types of studies often do not include statistical analyses, as shown in several classical studies for the past two decades, references as (1–4)

Furthermore, the most important parameter regarding the expression of genes in each stage of cell differentiation progress is their “relative” expression. As shown in all the gene expression data (Figures 2, 3, 6 and 7), we presented data as “Relative” gene expression compared to the baseline expression in the undifferentiated P19 cells. 

Figure. 5 shows ELISA data. Due to technical limitation in detecting thyroglobulin secreted from limited cell numbers in the in vitro culture system, ELISA was carried to detect TG secreted (to medium) from cells pooled from three independent trials.

References:

  1. Zhang SC, Wernig M, Duncan ID, Brüstle O, Thomson JA. In vitro differentiation of transplantable neural precursors from human embryonic stem cells. Nat Biotechnol 19: 1129–1133, 2001. doi: 10.1038/nbt1201-1129.
  2. Lin R-Y, Kubo A, Keller GM, Davies TF. Committing Embryonic Stem Cells to Differentiate into Thyrocyte-Like Cells in Vitro. Endocrinology 144: 2644–2649, 2003. doi: 10.1210/en.2002-0122.
  3. Khoruzhenko A, Miot F, Massart C, Van Sande J, Dumont JE, Beauwens R, Boom A. Functional model of rat thyroid follicles cultured in Matrigel. Endocr Connect 10: 570–578, 2021. doi: 10.1530/EC-21-0169.
  4. Pandya H, Shen MJ, Ichikawa DM, Sedlock AB, Choi Y, Johnson KR, Kim G, Brown MA, Elkahloun AG, Maric D, Sweeney CL, Gossa S, Malech HL, McGavern DB, Park JK. Differentiation of human and murine induced pluripotent stem cells to microglia-like cells. Nat Neurosci 20: 753–759, 2017. doi: 10.1038/nn.4534.

#3. In Figures 4 and 7, immunofluorescence staining (IF) of thyroglobulin (TG) and DAPI is shown, but it seems that the gain is set high in the P19 cells as compared with that in primary thyrocytes, and the fluorescence is close to saturation. It should be imaged under appropriate conditions. In addition, the fluorescence intensity of IF varies greatly depending on the setting; therefore, it is not quantitative. If the authors want to compare the amounts in the histological specimens, they should use immunohistochemistry with DAB.

      Reply- Thank you for pointing this out. All immunofluorescence images were acquired under the same condition. P19-differentiated thyrocytes (Fig 4A left side panels) typically form smaller follicles compared to primary follicles (Fig. 4A right side panels). To quantify the strength of these signals, we obtained heatmaps/histograms of these images as shown in new supplemental figures S2, S3. The heatmaps/histograms show signal intensity (pixel values).

As shown in the histograms of P19-thyrocytes (Fig. S2, middle two sets, which were from Fig. 4A the middle, upper and lower images of the left panel), we could not detect any pixels reaching saturation, indicating signals are not saturating for these P19-thyrocytes. Clearly, heat maps do localize areas of stronger intensity, which were from the DAPI stain. The Fig. 4A right side panels (primary follicles) show stronger green fluorescence (indicating TG). The heatmaps of these panels, Fig. S2 the far right panel, indeed shows a much stronger thyroglobulin intensity, but not saturating.   

Similarly, heatmaps/histograms were obtained for Fig. 7B, and the new data were presented in new Fig. S3. While the heatmaps show stronger DAPI intensity (nuclear staining), it is not saturating.

Overall, based on heatmaps/histograms, the presented images are not really saturating.

#4. As shown in Figure 5, P19 cells differentiated into thyrocytes secreted less thyroglobulin in response to TSH than primary cultured thyrocytes. Thyroglobulin production is a fundamental function of the thyroid follicular cells. Low response to TSH stimulation is a major problem when using P19 cells as a model of thyroid follicular cells.

      Reply - Using this new procedure, we have demonstrated that  P19-differentiated thyrocytes can be functional, in terms of physiological response to stimulation. Most importantly, this procedure is reliable and much more cost- and labor-efficient, and therefore it can provide a new tool appropriate for molecular studies. We do recognize that the P19-differentiated thyrocytes are not as efficient as primary thyrocytes, and there are still rooms to improve (added in Discussion, lines 492-4.

#5. The experiments in Figures 6 and 7 show that the negative control siRNA served as the negative control. This negative control siRNA has been widely used in other studies and has almost no value. It is necessary to compare its function with that of siRNAs for certain thyroid-related genes such as TTF1 and PAX8.

      Reply - We utilized the most common negative control siRNA for a simple reason that it is the most common and reliable reagent used in numerous cell model systems including various stem/progenitor cell types to avoid unexpected biological problems. Also, there should be minimal technical issues in terms of knockout efficiency. The purpose of this experiment is to demonstrate that the use of this reagent and the transfection procedure did not significantly disturb the progress of P19-thyrocyte differentiation. This would justify the application of P19 system in future applications, expecially using a gene knockout procedure, for studying various genes that can affect thyrocyte differentiation process.  

#6. “Embryo body” should be “embryoid body.”

      Reply – The description was corrected to “embryoid body ” throughout the text (lines 14, 15, 62, 93, 98, 100, 214, 357) .

Reviewer 3 Report

Comments and Suggestions for Authors

The experimental work impressively shows the use of P19 stem cell cultures to generate thyrocytes with signs of maturation and generation of follicular structures. The follicles are able to secrete thyroglobulin and respond to TSH. Application tests revealed a robust culture model.

Comments:

Line 253: The formation of the follicle centers is unclear. Do the lumina of the follicles arise through the destruction/ apoptosis of the central cells or through rearrangement?

Line 321: Bold type of “were” is not necessary.

Line 336: Bold type of “control” is not necessary.

Line 337: These statements are more of a discussion than a result and should be moved to the Discussion part. It should also be discussed whether the follicles are neoplastic-autonomous (follicular carcinoma) or rather parenchymatous.

Author Response

Reviewer (3)

Line 253: The formation of the follicle centers is unclear. Do the lumina of the follicles arise through the destruction/ apoptosis of the central cells or through rearrangement?

      Reply - This sentence was rephrased as “Follicles form through thyrocytes rearrangement so that the apical surface of polarized thyrocyte faces the center, the colloid” line 251-252.

Line 321: Bold type of “were” is not necessary.

      Reply – It was corrected in line 319.

Line 336: Bold type of “control” is not necessary.

      Reply – Bold was removed, “control”, in line 334.

Line 337: These statements are more of a discussion than a result and should be moved to the Discussion part. It should also be discussed whether the follicles are neoplastic-autonomous (follicular carcinoma) or rather parenchymatous.

        Reply - The reviewer’s point is appreciated; this section was moved to “Discussion”, lines 485-494. We must emphasize that this work established a new procedure to induce P19 embryonal carcinoma cells to differentiate into functional thyrocytes that show a proper physiological response to stimulation. Even in the context of monolayer culture, the differentiated cells do polarize which would advance and rearrange to form “follicle-like” structures. Our work did not aim to study follicular thyroid carcinomas.

Round 2

Reviewer 2 Report

Comments and Suggestions for Authors

As for my previous comment #2, it is important to confirm reproducibility through multiple experiments in the field of stem cell research too. Stem cell research is also a part of biomedical research. I think it is inappropriate to say that repetition of the experiments is not necessary.

Comments on the Quality of English Language

I do not have any specific comments on the quality of the English Language.

Author Response

As for my previous comment #2, it is important to confirm reproducibility through multiple experiments in the field of stem cell research too. Stem cell research is also a part of biomedical research. I think it is inappropriate to say that repetition of the experiments is not necessary.

Reply- As commented by reviewer 2, we have modified several figures to show the reproducibility and to include the mean and standard error (SE) bars.
In revised Figure 2, we modified the Y-axis to show the folds of change from the baseline status (P19-undifferentiated) from three sets of experiments, instead of showing the levels of expression.
In revised Figure 3, we included three independent experiments to obtain the mean ± SE.
In revised Figure 6, we included three sets of data to obtain the mean ± SE, and modified the Y-axis to be consistent with that of Figure 2.
In Figure 7A, instead of showing one representative set of experiment as presented in our original figure, the revised figure 7A now shows the results of three independent sets of experiments at four different time points for easy comparison to Figure 3A.
The last paragraph of each figure legend was modified accordingly.

Round 3

Reviewer 2 Report

Comments and Suggestions for Authors

The current version of the manuscript looks much better.

My concerns are addressed.

Comments on the Quality of English Language

Nothing particular.

Author Response

Thank you very much